

# The honest sound of physical effort

Andrey Anikin

Cognitive Science, Lund University, Lund, Sweden

## ABSTRACT

Acoustic correlates of physical effort are still poorly understood, even though effort is vocally communicated in a variety of contexts with crucial fitness consequences, including both confrontational and reproductive social interactions. In this study 33 lay participants spoke during a brief, but intense isometric hold (L-sit), first without any voice-related instructions, and then asked either to conceal their effort or to imitate it without actually performing the exercise. Listeners in two perceptual experiments then rated 383 recordings on perceived level of effort ($n = 39$ listeners) or categorized them as relaxed speech, actual effort, pretended effort, or concealed effort ($n = 102$ listeners). As expected, vocal effort increased compared to baseline, but the accompanying acoustic changes (increased loudness, pitch, and tense voice quality) were under voluntary control, so that they could be largely suppressed or imitated at will. In contrast, vocal tremor at approximately 10 Hz was most pronounced under actual load, and its experimental addition to relaxed baseline recordings created the impression of concealed effort. In sum, a brief episode of intense physical effort causes pronounced vocal changes, some of which are difficult to control. Listeners can thus estimate the true level of exertion, whether to judge the condition of their opponent in a fight or to monitor a partner's investment into cooperative physical activities.

## INTRODUCTION

A major function of communication in the animal world is to regulate conflicts, maintain social dominance hierarchies, and signal physical fitness (*Bradbury & Vehrencamp, 1998*; *Maynard Smith & Harper, 2003*). Likewise, human social interactions are rife with demonstrations of fitness, dominance, and social prestige (*Chen Zeng, Cheng & Henrich, 2022*). From an evolutionary perspective, honest signals of fitness are adaptive in the context of male competition because they provide an opportunity to evaluate the formidability and motivation of potential opponents and to avoid costly fights (*Maynard Smith & Harper, 2003*). Even when it comes to actual blows, however, opponents continue to monitor each other's behavior for signs of weakness or indecision, attempting to gauge the level of each other's physical exertion and fatigue. In humans, such competitive interactions include numerous contact sports (*e.g.*, wrestling), not to mention actual fights. Physical effort is also routinely communicated vocally in non-confrontational contexts such as cooperative manual labor, sex, and childbirth (*Anikin & Persson, 2017*). Yet, very little is known about how effort is communicated vocally, and nothing at all about how far it is possible to exaggerate or conceal physical effort. Therefore, in this study I analyzed vocal signatures

Corresponding author
Andrey Anikin, rty_anik@yahoo.com

of actual, pretended, or concealed physical effort in speech produced during a brief, but intense and full-body muscular activity (gymnastic L-sit hold), looking for the acoustic changes that people have good control over, as well as for those that are harder to suppress or exaggerate and that could therefore constitute reliable indices of physical effort.

Most research on vocalizing under effort has focused either on stress rather than physical exertion *per se* (*Giddens et al., 2013*), or else on the risk of voice disorders in people whose profession demands regularly speaking during rigorous physical activity, such as athletic coaches and cheer-leaders (*Primov-Fever et al., 2013*; *Sandage, Connor & Pascoe, 2013*; *Van Puyvelde et al., 2018*). Accordingly, participants are usually recorded during emotionally stressful activities (*Johnstone & Scherer, 1999*; *Kirchhübel, Howard & Stedmon, 2011*; *Giddens et al., 2013*; *Van Puyvelde et al., 2018*) or prolonged physical exercise—for example, running on a treadmill (*Trouvain & Truong, 2015*) or pedaling on a stationary bicycle (*Baker, Hipp & Alessio, 2008*; *Sandage, Connor & Pascoe, 2013*; *Weston, Fuchs & Rochet-Capellan, 2020*; *Serré et al., 2022*). Not surprisingly, the most consistently reported vocal changes under these conditions include rapid and irregular breathing, which disrupts normal speech flow (*Baker, Hipp & Alessio, 2008*; *Fuchs, Reichel & Rochet-Capellan, 2015*; *Trouvain & Truong, 2015*), as well as elevated voice pitch and intensity (*Johannes et al., 2007*; *Kirchhübel, Howard & Stedmon, 2011*; *Primov-Fever et al., 2013*; *Fuchs, Reichel & Rochet-Capellan, 2015*; *Van Puyvelde et al., 2018*; *Weston, Fuchs & Rochet-Capellan, 2020*; *Emanuel & Ravreby, 2022*). Rapid perturbations in pitch and intensity (jitter and shimmer, respectively) are sometimes also reported to increase under effort (*Primov-Fever et al., 2013*), while other studies find no change (*Orlikoff, 2008*) or argue that these perturbations should actually decrease (*Van Puyvelde et al., 2018*). Overall, vocal changes under physical exertion are consistent with an increase in vocal effort, defined as the vocal features that vary with increasing communication distance (*Traunmüller & Eriksson, 2000*). Importantly, greater vocal effort results not only in higher loudness and pitch, but also in a change in voice quality, greatly increasing the amount of energy in the harmonics in a shift from breathy to tense or pressed phonation (*Johnstone & Scherer, 1999*; *Traunmüller & Eriksson, 2000*).

It is not known whether brief, but intense physical effort causes the same changes in the voice as does prolonged aerobic exercise, but some parallels are likely to exist, particularly considering that the most pronounced vocal changes are reported close to failure, as participants become exhausted (*Johannes et al., 2007*). In one of the few studies that involved lifting weights rather than exercising aerobically (*Orlikoff, 2008*), 20 speakers said /pi/or phonated continuously while holding weights with straight arms, which led to an increase in pitch variability and driving pressure, but no change in mean pitch, jitter, or phonatory airflow. In another recent study (*Emanuel & Ravreby, 2022*), in which participants lifted increasingly heavy weights, subjective ratings of effort by the speakers correlated with low pitch variability, as well as elevated average pitch and maximum intensity. Of note, speech was produced after each repetition rather than under load. Taken together, these studies suggest that the voice tends to become louder, more high-pitched, and pressed during physical effort, whereas the evidence is more mixed
regarding intonation and vocal instability. But are these honest signals of physical effort, or can they be manipulated at will by the vocalizer?

In some contexts it is obviously in the vocalizer's interest to suppress any cues to effort. For instance, opponents in a fight or sports match are motivated to conceal their exertions and fatigue, projecting the impression of untapped strength reserves. On cooperative occasions, in contrast, effort can be intentionally exaggerated, as in a game of tug-of-war played by two teams, where every player is eager to prove they are literally "pulling their weight". In the absence of direct evidence of what cues to physical effort in the voice are amenable to voluntary control, we can only speculate as to how this exaggeration or concealment might be achieved. However, it is important to emphasize that suppressing sounds of effort may incur a cost. The trunk and shoulder girdle need to be stabilized during intense effort by means of increasing the pressure in the lungs (*Orlikoff, 2008*; *Pouw & Fuchs, 2022*), which requires constricting the airways somewhere, and the larynx is anatomically well suited for the task. The opening between the vocal folds, known as the glottis, can be closed without phonating, as in glottal stops or unvoiced grunts, but bringing the vocal folds together increases the pressure in the lungs while maintaining respiratory airflow. Forced expiration through a closed glottis is used by weight lifters (*Tsai et al., 2010*) as well as hopping rodents, whose lungs absorb the shock of impact (*Blumberg, 1992*). As a result, rodents produce ultrasonic vocalizations during their jumps, while weightlifters emit loud grunts (*Anikin & Persson, 2017*). There is evidence that such grunting actually improves performance, as seen from the increase in throwing velocities in ball sports (*O'Connell et al., 2014*; *Tammany et al., 2021*). A person wishing to conceal vocal cues to effort can either hold their breath, which cannot be sustained given the high metabolic demands during physical activity, or keep the airways wide open so as to avoid wheezing or vocalizing with a pressed voice quality. The latter strategy may work well during mild aerobic exercise, but fully opening the glottis sacrifices the benefits of elevated lung pressure for bodily integrity during intense effort.

If effort needs to be concealed despite this potential cost, the vocalizer will need to prevent subglottal pressure and pitch from rising and control vocal effort, avoiding pressed phonation. Inversely, vocal exaggeration of physical effort can presumably be achieved by tensing the core and laryngeal muscles. The question is whether some aspects of phonation under physical effort are harder to suppress or to imitate in the absence of an actual load on the muscles, and vocal instability or tremor is one possible candidate. Increased muscular tension under intense physical activity may disturb phonation due to accidental activation of laryngeal musculature as well as intense and irregular airflow (*Primov-Fever et al., 2013*), so unstable phonation in general, and occasional rapid tremor in particular, could be regarded as a side effect of intense physical effort, particularly during full-body muscular activities accompanied by jerky movements and shaking. On the other hand, acoustic lie detectors commonly assume that the 8–14 Hz microtremor in laryngeal muscles disappears under stress, although recent reviews have concluded that there is very little evidence to support this assumption, and it is not clear whether the 10 Hz vocal microtremor even exists in laryngeal muscles, let alone how it changes under stress (*Kirchhübel, Howard & Stedmon, 2011*; *Giddens et al., 2013*; *Van Puyvelde et al., 2018*). Furthermore, there are

reports of flattened intonation during effort (*Emanuel & Ravreby, 2022*), leading some theorists to suggest that top-down regulation of vocal production is responsible for a flattening of natural pitch variation under emotional or physical stress (*Van Puyvelde et al., 2018*).

There are thus conflicting opinions about the effect of stress or effort on pitch perturbations, but it is important to distinguish between different eliciting contexts and time scales on which these perturbations occur. Speech intonation corresponds to slow, controlled pitch modulation, whereas jitter and shimmer represent accidental variability across individual glottal cycles—thus, at about 100 Hz and above. In between these two extremes, perturbations at 2–5 Hz (*Schoentgen, 2002*) or 8–12 Hz (*Titze, 2000*), known as vocal tremor or microtremor, occupy roughly the same frequency range as the vibrato, which is typically produced at 4–7 Hz in Western music—anything faster than that becomes too rapid for voluntary control. In principle, the changes at these time scales could be independent, caused by different physiological or neurological mechanisms, and therefore associated with different activities.

In sum, extrapolating from the better studied aerobic exercise, intense effort of the kind exerted during a fight can be expected to disrupt the natural speech flow and to be accompanied by an increase in vocal effort (higher voice intensity and pitch) and a shift to pressed phonation with strong harmonics. Two opposing predictions can be made with regard to vocal tremor, which is particularly interesting because it may be harder to control. Therefore, in this study vocal correlates of effort were compared in natural, concealed, or pretended physical exertion, and vocal instability at approximately 10 Hz was manipulated experimentally to investigate its possible role as an honest marker of physical effort. The original and tremor-manipulated recordings were also tested in two perceptual experiments, in which listeners either rated the apparent level of effort or classified the recordings categorically as relaxed speech, actual effort, pretended effort, or concealed effort. The results confirmed that physical exertion was accompanied by acoustic changes broadly indicative of increased vocal effort, namely elevated voice intensity, pitch, and a pressed voice quality. Interestingly, vocal tremor proved more difficult for speakers to either suppress or imitate, and its addition produced the impression of imperfectly concealed effort.

## MATERIALS & METHODS

### Recordings

The recordings ($N = 319$) were obtained from 33 speakers (15 females + 18 males, age mean $\pm$ SD = 28 $\pm$ 8), who were recruited among students and junior staff members at Lund University and simultaneously participated in another voice production experiment. The same neutral phrase (*Where were you a year ago?*) was recorded in four conditions: *relaxed* phonation (baseline), *real effort*, *concealed effort*, and *pretended effort* (Fig. 1). The *real effort* task was to lift straight legs as high as possible while seated in a chair with arm support. This simplified version of the full L-sit or V-sit position powerfully activates the core and hip flexors, but does not require as much compression or straight arm strength,

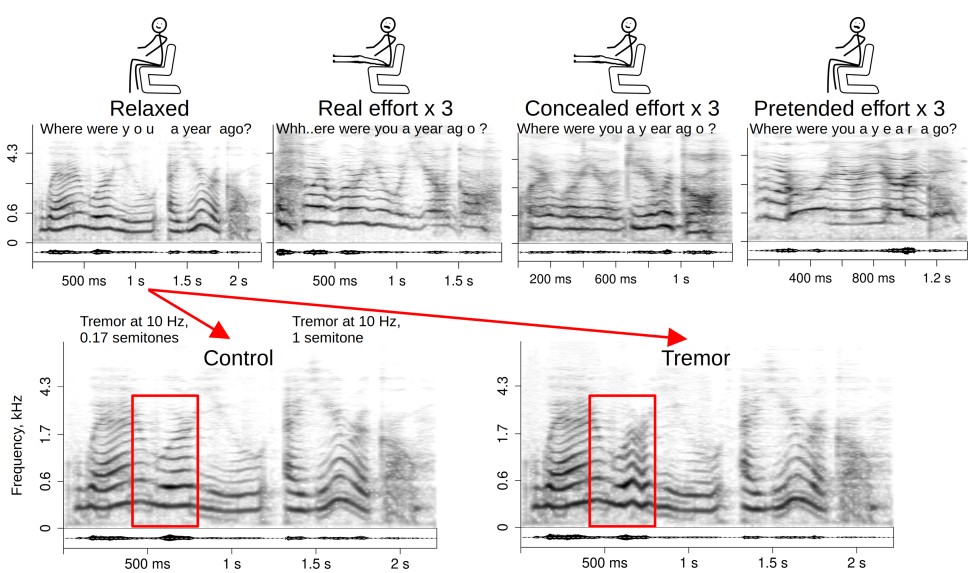

**Figure 1** **Stimuli and manipulations.** Recording conditions (above) and the manipulation intended to mimic vocal tremor (below) by speaker "cag" (audio in supplements). The shown example of *tremor* manipulation was among the more noticeable perceptually and had the strongest effect on the ratings of perceived effort compared to other stimuli. Spectrograms with a 50 ms Gaussian window and frequency on a bark scale.

making it suitable for the non-athletic population. Participants were explicitly instructed to engage the core, and an experimenter demonstrated the correct position. This static hold was maintained just long enough for participants to say, "*Where were you a year ago?*" The *concealed effort* task was identical, but now the participant was asked to sound as normal as possible during the hold—that is, to conceal the physical effort. Finally, for the *pretended effort* task the participant remained seated in a relaxed position, but pretended to be making the same kind of physical effort. Each task was repeated three times with brief pauses in between in a relaxed position, and each take was saved as a separate audio file.

The voice was recorded with two channels: a tie clip microphone connected to a laptop and a Tascam DR-05 recorder with 0 dB gain at approximately 1.5 m from the speaker. Out of 319 recordings, 149 relatively quiet utterances were saved from the tie clip microphone, while 170 louder utterances were saved from Tascam to avoid clipping. Because the channels were individually calibrated for each participant, it was possible to calculate the approximate sound pressure level (dB-C at 1 m) for each recording, except for one speaker who was mistakenly recorded only with the clip microphone. Ethical approval for performing experiments with human subjects was provided by the Comité d'Ethique du CHU de Saint-Etienne (IRBN692019/CHUSTE). Participants signed an informed consent form before the recording session and were debriefed afterwards; there was no formal payment.

## Acoustic analysis

All vocalizations were analyzed acoustically with the functions *analyze* and *pitchDescriptives* from R package *soundgen* 2.5.1 (*Anikin, 2019*) using pre-extracted, manually corrected (in the *pitch_app* interactive environment) contours of the fundamental frequency. Pitch values were saved every 25 ms, which means that frequency modulation could be analyzed at frequencies up to 20 Hz. In actual fact, the target range for frequency modulation was analyzed in the range from 8 to 14 Hz, so as to focus specifically on the region commonly associated with vocal tremor (*Giddens et al., 2013*). The first four resonance frequencies (formants F1–F4) were extracted using linear predictive coding with manual verification in *formant_app*, averaging over entire utterances, but only F1 was analyzed further because of its expected association with mouth opening and loudness (*Titze, 2000*).

It was also important to analyze spectral correlates of vocal effort: loudness and its variability (root mean square amplitude and its inter-quartile range); pitch and its variability (median and inter-quartile range); energy in harmonics (the proportion of spectrum above 1.5 times the fundamental); height of harmonics (frequency to which harmonics remain detectable in the spectrum); first spectral quartile (25th percentile of the spectrum); and spectral centroid (center of gravity of the spectrum). Other acoustic measures included duration (inversely proportional to speech rate because all speakers said the same sentence); pitch slope (mean absolute pitch slope in semitones); and the number of rapid and slow pitch inflections per second (calculated after low-pass filtering the intonation contours at 10 Hz and 1 Hz, respectively). A few more acoustic descriptives were considered in a more exploratory fashion: spectral novelty (a measure of spectrotemporal variability estimated from a self-similarity matrix); cepstral peak prominence (a measure of pitch quality, the ratio of the highest peak in the cepstrum to the regression line drawn through the ceptsrum); harmonics-to-noise ratio (maximum autocorrelation—a measure of tonality); roughness (proportion of the modulation spectrum within the roughness range of frequencies); and apparent vocal tract length (estimated from formants F1–F4). Formant-related measures, loudness, spectral novelty, and roughness were analyzed in all frames, while the rest of spectral descriptives and pitch-related measures were only analyzed in voiced frames.

## Stimulus manipulation

To test the hypothesis that vocal instability or tremor contributes to the perception of physical effort, baseline recordings of a relaxed voice were resynthesized adding a small amount of tremor with the function *shiftPitch* in *soundgen* package (*Anikin, 2019*), which dynamically adjusts the fundamental frequency with a phase vocoder, while preserving speech rate and formant frequencies. A tremor frequency of 10 Hz was chosen to correspond to the typical rate of frequency modulation (FM) detected in natural utterances with audible vocal instability. To approximate a tremulous voice of this type, Gaussian noise was added to the original $f_o$ with 20 anchor points per second of audio. The SD of this Gaussian distribution was set to 1 semitone (~8.3%), which was audible but not unnaturally strong. The tremor was not applied over the entire duration of the recording, but its amplitude was modulated over time with a Gaussian filter spanning 50% of the sound's duration, so that the strongest tremor occurred relatively briefly, timed to coincide with voiced, relatively

loud sections (Fig. 1). When needed, the resynthesis script was rerun several times to ensure that the tremor was perceptually detectable and sounded as natural as possible. To ascertain that the perceptual effects of the manipulation were not a mere side effect of resynthesis, a *control* condition was created by adding the same type of tremor, but with time-uniform and very small amplitude (SD = 1%, or 0.17 semitones). Acoustic analysis (below) confirmed that the average depth of frequency modulation increased considerably in the *tremor* condition (+0.11 (95% CI [0.06, 0.18], reaching approximately the same level as in unmanipulated recordings of real effort), while it increased only marginally (+0.05 [−0.01, 0.10]) in the *control* condition.

## Perceptual tests

Two perceptual experiments were written in javascript and conducted online: a rating study for testing the perceived level of physical effort and a forced-choice categorization study. In both cases, listeners were presented with a random selection of 99 out of the total of 383 experimental stimuli (319 original recordings and 64 manipulated recordings). In the rating study, they were asked to rate "*How much physical effort is this person making?*" on a horizontal analog magnitude scale labeled from *Very light effort* to *Extreme effort*. In the forced-choice study, listeners were asked to pick among four categories: *Relaxed*, *Concealed effort*, *Real effort*, or *Pretended effort*. The sounds could be replayed as many times as needed, and there was no time limit for responding.

## Listeners

All participants ($N = 141$, 73 females, age 18 to 69, mean 30.9) were recruited on http://prolific.co/ and paid for their time. Every participant who performed at least 20 trials was included in the analysis ($n = 39$ participants in the rating study and 102 in the forced-choice study). The sample sizes were chosen so as to provide adequate precision of the Bayesian estimates of effect sizes, corresponding to 10.1 responses per stimulus in the rating study and 26.4 in the forced-choice study. Participants were informed about the goals of the study and provided informed consent by agreeing to the conditions at the beginning of online experiments.

## Data analysis

Data was analyzed with Bayesian multilevel models using the R package *brms* 2.18.0 (*Bürkner, 2017*) with mildly informative conservative priors. All estimates shown in the text are medians of posterior distributions and 95% credible intervals. Unbounded continuous variables, such as pitch or loudness, were analyzed with Gaussian models, bounded ratings from perceptual experiments with beta models (rescaled to range from 0.01 to 0.99), and categorical outcomes with multi-logistic regression. Considering the large number of acoustic predictors ($n = 19$) relative to the number of stimuli to classify ($n = 319$), the confusion matrix for acoustic classification of experimental condition was obtained with Random Forest, a simple and robust algorithm for such situations (*Breiman, 2001*). Random Forest builds multiple decision trees and has the advantage over multiple regression in being robust to collinearity and capable of capturing complex interactions between predictors; it is also convenient to use because out-of-sample classification error

and variable importance are estimated internally. Acoustic variables were normalized to have a mean of 0 and SD of 1, and frequency measures originally expressed in Hz were log-transformed prior to normalization. More details on the structure of each model are given in figure legends; the code and datasets for reproducing all analyses, as well as the audio recordings, can be downloaded from https://osf.io/pxdhq/.

# RESULTS

## Acoustic signatures of effort

When speakers said "*Where were you a year ago?*" during real, pretended, or concealed physical effort, a number of acoustic changes occurred compared to saying the same phrase in a relaxed state. These changes were quantified with 19 spectral descriptives (Fig. 2A). Both $f_o$ and measures of vocal effort, such as loudness, energy in harmonics, and the first spectral quartile, were noticeably elevated in all three *effort* conditions. For example, $f_o$ was 0.25 SD (95% CI [0.08, 0.42]) higher in the *concealed effort* condition compared to baseline, 0.44 SD [0.26, 0.62] higher in the *real effort* condition, and 0.72 SD [0.55, 0.89] higher in the *pretended effort* condition (all effect sizes with 95% CIs are shown in Fig. 2A). Some acoustic changes were shared primarily by the *real* and *pretended effort* conditions and were less pronounced in the *concealed effort* condition, indicating that these aspects of voice production were voluntarily exaggerated or suppressed, respectively, to create an impression of physical effort or to hide it. For instance, the energy in harmonics and their height (the maximum frequency to which harmonics of $f_o$ remain detectable in the spectrum) were elevated in the *real effort* and *pretended effort* conditions, but not in the *concealed effort* condition, which also showed a smaller increase in voice pitch (0.25 SD above baseline, 95% CI [0.08, 0.42]) than in the *real* (+0.44 SD [0.26, 0.62]) and *pretended effort* conditions (+0.72 SD [0.55, 0.89]). These acoustic changes are broadly consistent with pressed phonation under physical effort, and speakers are clearly both aware of this change in voice quality and able to manipulate it in order to exaggerate or conceal their effort.

Revealingly, there were also some acoustic changes that were more pronounced in conditions associated with actual, but not pretended physical effort. Speech rate increased in the *real effort* condition (duration −0.42 SD [−0.75, −0.09]), and even more so during *concealed effort* (−0.54 SD [−0.85, −0.24]), but not *pretended effort* (+0.26 SD [−0.06, 0.57]). An interesting feature was the increase in the median depth of frequency modulation in the 8–14 Hz range, particularly noticeable during *real effort* (+0.54 SD [0.11, 0.94]) and *concealed effort* (+0.76 SD [0.37, 1.13]). It is difficult to capture short episodes of vocal tremor with time-averaged measures, but they appear to be perceptually salient, and their role was confirmed experimentally (next section).

A related question is how well the recording condition can be predicted from the combination of measured acoustic characteristics. A Random Forest classifier achieved an average recognition accuracy of 49% (95% CI [39, 59]) across four production categories using the same 19 acoustic descriptors. The *relaxed* category was recognized more accurately (64% [36, 90]) than the *real effort*, *concealed effort*, and *pretended effort* categories (45% [21,

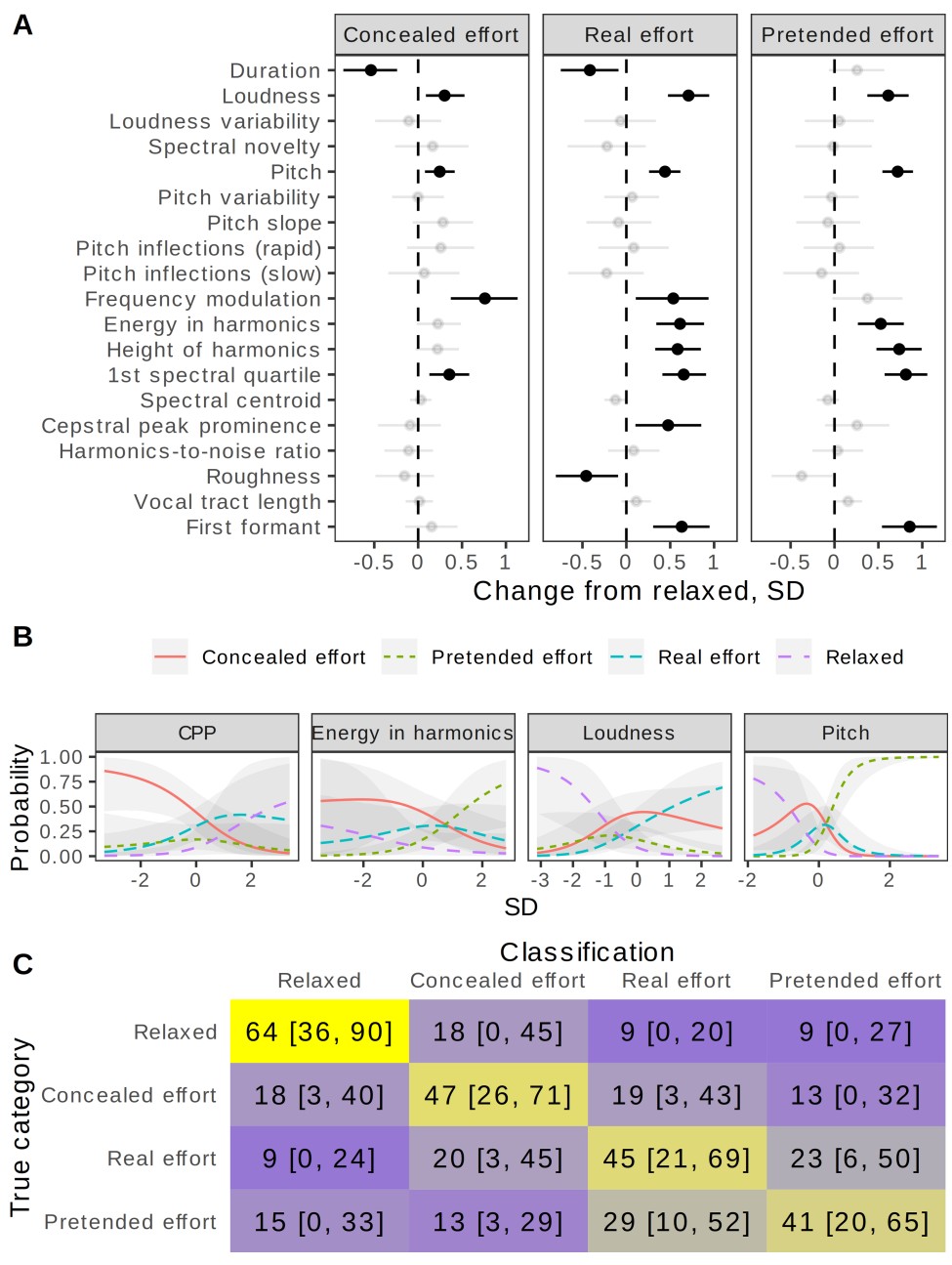

Figure 2 **Voice production: acoustic correlates of actual physical effort.** (A) Some acoustic changes are shared by real and pretended effort, indicating that they are under voluntary control, whereas others are shared by real and concealed effort, suggesting that they are difficult to suppress. Multivariate Gaussian mixed model predicting change in 19 vocal features as a function of condition after controlling for speaker sex and recording channel, with a random intercept per speaker. Shown: difference between each condition and baseline, in SD (median of posterior distribution and 95% CI), grayed out if <99% of posterior distribution was either positive or negative. (B) Independent predictors of physical effort include loudness, pitch, and measures of voice quality. (continued on next page...)

**Figure 2 (…continued)**
Categorical multiple regression predicting condition from 19 acoustic characteristics and speaker sex; only predictors with a robust effect of at least 25% on at least one category are shown. X = standardized value of an acoustic predictor holding all other predictors at their mean values; Y = fitted probability of each outcome category (median of posterior distribution and 95% CI). For example, other things being equal, the probability of classifying a recording as *pretended effort* sharply increases with its pitch and energy in harmonics. (C) Confusion matrix of a Random Forest model for predicting condition from 19 acoustic characteristics, shown as the median (% per category) and 95% coverage interval from 1000 simulations. Gray = chance level (25%), blue = below chance, yellow = above chance. $N = 319$ recordings.

69], 47% [26, 71], and 41% [20, 65], respectively). Looking at the confusion matrix (Fig. 2C), *pretended effort* most closely resembled *real effort*, whereas *concealed effort* was easily confused with either *real effort* or *relaxed* conditions. The acoustic predictors of condition were similar to the acoustic variables showing the greatest change from baseline (Fig. 2A). When all 19 were considered simultaneously in multiple regression, the most important predictors of condition were loudness, pitch, energy in harmonics, and cepstral peak prominence—a correlate of breathy voice quality, which unexpectedly distinguished the *concealed effort* condition after controlling for the remaining acoustic predictors (Fig. 2B).

## Perception of effort in the voice

Listeners in two perceptual experiments were presented with the original recordings and modified versions with some vocal tremor added at a minimal level in the *control* condition and at a perceptually noticeable level in the *tremor* condition. The task was either to rate the perceived level of effort on a continuous scale (Fig. 3) or to classify the recordings into four conditions (Fig. 4).

Effort ratings increased by 0.19 on a scale of 0 to 1 (95% CI [0.13, 0.26]) compared to the *relaxed* condition when speakers made a *real effort*, and this change was even more pronounced in the *pretended effort* condition (+0.33 [0.25, 0.41]), indicating that these "fake" expressions of effort were exaggerated, or perhaps stereotypical and therefore easily recognizable (Fig. 3B). Interestingly, speakers failed to completely conceal their effort: effort ratings in the *concealed effort* condition were still considerably above baseline (+0.10 [0.06, 0.14]), if also below the level of *real effort* (−0.09 [−0.15, −0.04]).

Adding vocal *tremor* to recordings made in the *relaxed* condition produced a small, but statistically robust increase in the perceived level of effort (+0.08 [0.03, 0.13]). No change was observed in the *control* condition when the same manipulation was performed at very low levels (+0.01 [−0.03, 0.05]), confirming that the effect of *tremor* was not a mere side effect of audio resynthesis. The perceptual association of unstable phonation with effort was also captured indirectly by the positive effect of frequency modulation depth on effort ratings across all conditions (+0.02 for each SD, 95% CI [0.00, 0.03]) after controlling for speaker sex and other acoustic predictors with multiple regression (Fig. 3C). This effect was quite weak, however, and the main acoustic predictors of effort ratings were pitch (+0.16 [0.12, 0.21]), energy in harmonics (+0.05 [0.02, 0.07]), and duration (+0.04 [0.02, 0.05]). This is consistent with the analysis of vocal production above and indicates that listeners primarily attended to correlates of vocal effort and instability when judging the level of physical effort.

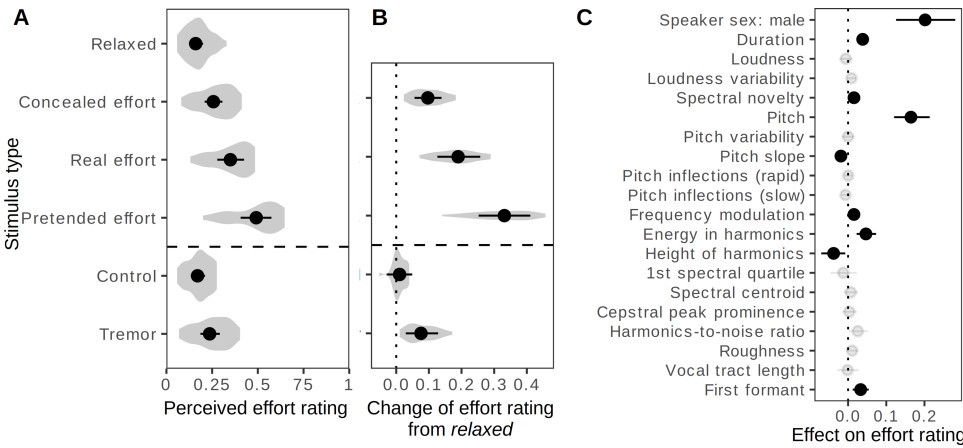

**Figure 3   Perception of effort in the voice: continuous ratings.** (A–B) Ratings of perceived effort increased when speakers engaged in real or pretended effort, or when a small amount of vocal instability was added to the *relaxed* recordings (*tremor*). Multilevel beta regression modeling ratings in individual trials as a function of recording condition, allowing the effect of condition to vary across speakers and assigning group-levels intercepts to each speaker and sound. Shown: fitted value per condition (A) and contrasts from baseline (B) as medians of posterior distribution and 95% CIs; violin plots show the distribution of effects across listeners. (C) Acoustic predictors of effort ratings included higher pitch and more energy in harmonics. Multilevel beta regression modeling effort ratings as a function of speaker sex and 19 acoustic variables with group-level intercepts for each speaker, listener, and sound. Shown: predicted change in effort rating, on a scale of 0 to 1, when changing one acoustic variable at time by 1 SD and keeping all other variables constant (median of posterior distribution and 95% CI), grayed out if <99% of posterior distribution was either positive or negative. $N = 3860$ ratings of 383 recordings by 39 listeners.

The forced-choice classification task provided additional details on how effort was detected by listeners. The four unmanipulated conditions were recognized correctly in 32.8% of trials (chance level = 25%), but the confusion was largely within the three effort conditions (Fig. 4A). The *relaxed* condition was recognized correctly 65% of the time (95% CI [59, 71]), and *pretended effort* was also recognized moderately well (41% [30, 52]), suggesting that it sounded slightly "fake". *Concealed effort* was very often (43% [38, 48]) misidentified as *relaxed* voice, suggesting that speakers managed to conceal most clues to physical effort. The most common response in both *control* and *tremor* conditions was *relaxed* (68% [61, 74] and 47% [41, 54], respectively), but the probability of selecting *relaxed* was 21% [14, 27] higher for the *control vs. tremor* condition. Curiously, the most likely other classification of *tremor* was *concealed effort* (27% [21, 35], or 12% [6, 19] higher than in the *control* condition). Thus, the addition of even minimal vocal tremor to originally relaxed utterances did not simply enhance the perceived level of effort, as suggested by the ratings above, but specifically created the impression of imperfectly concealed effort. Looking at the acoustic predictors of classifications across all conditions, slow speech rate and high pitch distinguished between the *relaxed* and *pretended effort* response categories, while an increase in frequency modulation made it less likely that a recording would be classified as *relaxed* (Fig. 4B).

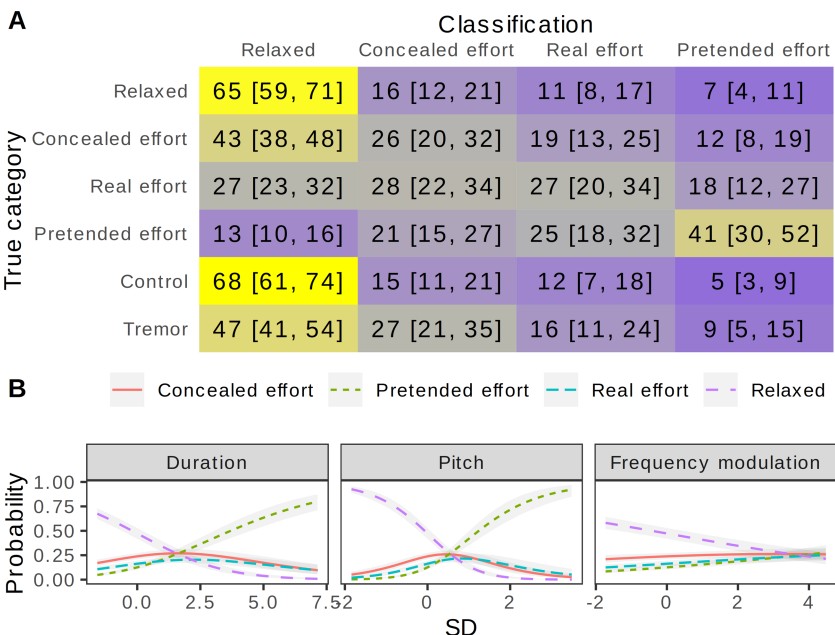

**Figure 4  Perception of effort in the voice: forced-choice classification.** (A) Confusion matrix of listeners' classifications, with proportions of responses within each true category shown in % (medians of posterior distributions and 95% CI). Categorical multilevel regression predicting the chosen response category as a function of recording condition, allowing the effect of condition to vary across listeners and with group-level intercepts for each speaker and recording. Gray = chance level (25%), blue = below chance, yellow = above chance. (B) Classification of recordings as a function of their acoustic characteristics. Predicted relative probabilities of the four response categories from a multilevel categorical model predicting response from speaker sex and 19 acoustic variables, with group-level intercepts for each listener. Shown: partial effects of normalized acoustic predictors that reliably changed the probability of any category by at least 25% over the observed range of each predictor (medians of posterior distributions and 95% CIs). N = 10099 classifications of 383 recordings by 102 listeners.

## DISCUSSION

A lot of research in evolutionary and social psychology has focused on displays of physical and social dominance in general (*Chen Zeng, Cheng & Henrich, 2022*) and on vocal correlates of dominance in particular, such as low voice pitch and formant frequencies (*Aung & Puts, 2020*). However, much less is known about the vocal signals exchanged once a confrontation has already escalated: how can opponents evaluate each other's condition and decide whether to withdraw from a fight? Vocal signs of physical effort may provide important cues. Likewise, physical effort needs to be monitored in cooperative efforts, games, and indeed in sexual contexts. Yet, it remains unclear how the voice changes during different kinds of physically stressful activities, and the crucial aspect of reliability or honesty (*Maynard Smith & Harper, 2003*) of vocal signals of physical effort has not been investigated. Accordingly, in this study I analyzed changes in the voice during real, pretended, or concealed physical effort and found several acoustic correlates of effort, of which high vocal effort (increased loudness, pitch, and a pressed voice quality) is the most salient, and vocal tremor the most reliable or honest.

The most obvious change was that the voice became louder, more high-pitched, and tense or pressed under intense effort. This is consistent with previous research on the effects of long-term aerobic exercise on the voice, particularly close to physical exhaustion (*Johannes et al., 2007*; *Kirchhübel, Howard & Stedmon, 2011*; *Van Puyvelde et al., 2018*). There is a clear physiological mechanism responsible for these changes: muscular activation in the upper body increases subglottal pressure, elevating both voice intensity and the frequency with which the vocal folds vibrate, perceived as voice pitch (*Titze, 2000*). These changes may not be a mere side effect of physical activity: it is physiologically advantageous to partly close the glottis and thus raise the pressure in the lungs during physical exertion (*Orlikoff, 2008*; *Pouw & Fuchs, 2022*), which may contribute towards the pressed or even wheezy voice quality that is so familiar from everyday situations and portrayals in the media (*e.g.*, the famous "number two" scene in Austin Powers: https://www.youtube.com/watch?v=nmJKY59NX8o).

The acoustic changes corresponding to increased vocal effort (increased intensity, pitch, and energy in harmonics) were broadly similar regardless of whether speakers actually performed the exercise or merely pretended to do so, although pretended effort was somewhat stereotypical and exaggerated, which is presumably why listeners often correctly identified it as such. Likewise, speakers were quite good at concealing their effort: voice loudness and pitch remained slightly elevated, but most of the acoustic signatures of effort were controlled successfully. The main exception was frequency modulation in the tremor range, here measured at 8–14 Hz. This type of pitch perturbation remained very pronounced in the *concealed effort* condition and predicted perceived effort ratings after accounting for other acoustic variables. Crucially, an illusion of concealed effort could be created by adding a brief, barely noticeable episode of vocal tremor to resynthesized baseline recordings made in a relaxed state, confirming that a trembling voice may be an honest and perceptually salient index of genuine effort.

There is conflicting evidence regarding the effect of stress and physical effort on pitch perturbations, including vocal microtremor (*Kirchhübel, Howard & Stedmon, 2011*; *Giddens et al., 2013*; *Van Puyvelde et al., 2018*), but the mechanism behind the trembling voice in this study is likely to be extralaryngeal and very simple, stemming from fluctuations in the recruitment of large motor units in major muscles of the upper body. As the load increases or the muscle fatigues, progressively larger motor units with higher activation thresholds are recruited in the spinal cord in accordance with Henneman's size principle (*Henneman, 1957*). Because all muscle fibers controlled by a single motor neuron are activated at once in an all-or-nothing manner, the recruitment or failure of these large motor units causes noticeable fluctuations in the force output, so that movements become less smooth and more jerky under heavy load. In an isometric hold, such as the L-sit performed by speakers in this study, this manifests in some shaking, which causes transitory perturbations in the vocal output because of fluctuations in the lung pressure, and probably also *via* biomechanical coupling with laryngeal musculature. This is another manifestation of the increasingly well-recognized principle that bodily movements affect the voice. For example, voice intensity and pitch show little perturbations caused by leg movements (*Serré et al., 2022*), or even by ordinary gestures that accompany speech (*Pouw & Fuchs, 2022*).

The interesting aspect of the kind of vocal tremor described here is that these perturbations may be a little too rapid for voluntary control, and therefore difficult not only to suppress, but also to fake convincingly, unless the speaker powerfully contracts the abdominal muscles—that is, essentially recreates physical effort without actually performing the exercise (which some speakers may have done in the *pretended effort* condition). In this sense, vocal tremor is similar to involuntary twitching of the eyelids or fingers (*Schoentgen, 2002*) or to physiological tremor in general, which occupies the same frequency range, peaking at about 9 Hz in humans (*Marshall & Walsh, 1956*). In future it will also be interesting to investigate the vocal tremor caused by other types of physical effort, as well as by shivering induced by hypothermia or extreme nervousness.

It is doubtful that vocal tremor evolved to signal stress or exertion: most likely, this is simply a physiological side effect of muscular activity, and thus a cue rather than a signal (*Maynard Smith & Harper, 2003*). It is particularly interesting that such indexical vocal cues are found in humans—a species whose flexibility of vocal control and capacity for vocal production learning are rivaled by songbirds, but few other mammalian species (*Vernes et al., 2021*). Vocal learners can sometimes escape from physiological constraints on voice production: for example, in-flight vocalizations tend to be synchronized with wing beats in most birds, but not in the species capable of vocal learning (*Berg, Delgado & Mata-Betancourt, 2019*). Like songbirds, humans have even evolved direct connections between the motor cortex and laryngeal motor neurons, which bypass the phylogenetically shared limbic pathway and enable fine voluntary control over vocal production (*Ackermann, Hage & Ziegler, 2014*); this neurological adaptation contrasts sharply with the relatively stereotypical and affectively triggered vocalizations in most mammals (*Owren, Amoss & Rendall, 2011*). Despite human vocal flexibility, however, listeners can often determine the authenticity of emotion expressed by speech (*Drolet, Schubotz & Fischer, 2012*) as well as nonverbal vocalizations (*Anikin & Lima, 2018*). How is this possible? Hard-to-fake acoustic characteristics of the human voice furnish important clues for distinguishing between authentic physiological or affective states and their volitional imitations, and physiologically determined "slips" caused by imperfect vocal control are particularly informative because of their very unpredictability. It is difficult to control nonlinear vocal phenomena, such as sudden frequency jumps or episodes of chaos (*Fitch, Neubauer & Herzel, 2002*), inspiratory or ingressive phonation in bouts of intense sobbing or laughing (*Anikin & Reby, 2022*), or vocal tremor. Accordingly, their appearance provides reliable information about the speaker's emotional or physiological state (*Berg, Delgado & Mata-Betancourt, 2019*).

The present study has several limitations. Since two different microphones were used, some of the spectral measures, such as harmonics-to-noise ratio and spectral centroid, may be affected by the choice of recording channel. This has no bearing on the main reported acoustic correlates of physical effort, including loudness, pitch, and tremor, but it would be advisable to standardize the recording conditions in future studies. The choice of physical activity may also have a major effect on the changes detected in the voice. The L-sit was chosen as an approximation to the kind of brief, full-body muscular effort that would occur in a fight-like context. This exercise has the advantage of being very easy to set up with no

special equipment and can challenge both untrained and athletic individuals, depending on the angle to which the legs are raised. However, the load is self-regulated, which can amplify the already considerable individual variability in vocal signatures of effort (*Johannes et al., 2007*; *Kirchhübel, Howard & Stedmon, 2011*; *Van Puyvelde et al., 2018*). Furthermore, unless the angle is measured from video recordings, it is difficult to quantify the performance and estimate whether the generated force or power output were reduced when speakers attempted to conceal their effort, which would be an interesting question for follow-up studies in light of the evidence that natural vocal expressions of effort help to maximize athletic performance (*O'Connell et al., 2014*; *Tammany et al., 2021*).

In an interesting recent methodological development, the technique of real-time pitch shifting (*Rachman et al., 2018*) was used to create the impression of a tremulous voice by adding a small amount of regular vibrato at 8 Hz to speech, singing, and instrumental or vocal music (*Bedoya et al., 2021*). The effects depended on the type of stimulus tested: the perceived valence usually became more negative, while arousal ratings increased in screams and decreased in other stimuli. While a steady vibrato is different from brief episodes of irregular tremor tested here, there is probably some perceptual overlap between these phenomena, and real-time manipulations of different types of perturbed phonation can be a promising avenue for future research. It will also be interesting to extend the investigation of effort concealment and exaggeration from brief bursts of effort of the kind examined in this study to steady aerobic exercise: both types of effort affect the voice, and both co-occur in numerous physical activities.

It is important to reiterate that physical confrontations or their proxies, such as athletic competitions, while particularly relevant from an evolutionary perspective, are far from being the only contexts in which vocal communication of effort occurs. Like other primates, humans are very vocal during copulations, at least in some cultures. Female sexual moans are attractive to males and appear to be produced voluntarily in order to please the partner (*Prokop, 2021*), but listeners judge these vocalizations to indicate not only pleasure, but also physical effort, and the same applies to groans of pain (*Anikin & Persson, 2017*). Many other cooperative activities also involve coordinated physical effort; in fact, one of the earliest theories of language evolution posited that speech itself arose in this context (*Fitch, 2010*). While this *heave-ho* theory is now merely a historical curiosity, verbal and nonverbal vocal expressions of effort are ubiquitous in everyday life, including contexts that have crucial consequences for inclusive fitness. It is therefore important to understand how effort is vocally communicated—or concealed.

## ACKNOWLEDGEMENTS

I would like to thank all 33 speakers who volunteered their time and effort for producing the vocal stimuli for this study. I am also grateful to Susanne Fuchs, Aviv Emanuel, and Wim Pouw for providing many constructive comments on the first draft of this paper.

### Funding
Andrey Anikin was supported by grant 2020-06352 from the Swedish Research Council (Vetenskapsrådet). The funders had no role in study design, data collection and analysis, decision to publish, or preparation of the manuscript.

### Grant Disclosures
The following grant information was disclosed by the author:
Swedish Research Council (Vetenskapsrådet): 2020-06352.

### Competing Interests
The author declares that he has no competing interests.

### Author Contributions
- Andrey Anikin conceived and designed the experiments, performed the experiments, analyzed the data, prepared figures and/or tables, authored or reviewed drafts of the article, and approved the final draft.

### Human Ethics
The following information was supplied relating to ethical approvals (i.e., approving body and any reference numbers):

Ethical approval for performing perceptual experiments with human subjects was provided by the Comité d'Ethique du CHU de Saint-Etienne (IRBN692019/CHUSTE).

### Data Availability
The audio files, datasets, and R scripts for statistical analysis are available at OSF: Anikin, Andrey. 2022. ''The Honest Sound of Effort.'' OSF. December 4. https://www.doi.org/10.17605/OSF.IO/PXDHQ.

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
