# Peer review of "The honest sound of physical effort"

_PeerJ, doi:10.7717/peerj.14944_

## Round 0.1 · original submission · Minor Revisions

Your manuscript has now been seen by 3 reviewers. You will see from their comments below that they find your work of interest, and some constructive points are worth considering. We therefore invite you to revise and resubmit your manuscript, taking into account the points raised. Please highlight all changes in the manuscript text file.

·

Basic reporting

Please make sure that readers get that tremors can be caused by different activities – a comparison of different frequencies may not be valid if different activities are compared
- Line 114: 8-14 Hz microtremor might be due to the muscular tremor of the body part that is activated in an isometric action – which is transferred to the larynx via the ribcage and the respiratory system. In un-isometric tasks, I don’t think such a tremor will be present. Rather, vocal folds may be stiffer (and f0 gets higher for this purpose). So, it really depends on the motor action.
- Line 127ff. While all tremors occur at the vocal fold level, they may be caused by different activities (as suggested in the Discussion – which I can wholeheartedly agree with). I don’t think that singing and vocalizing are comparable to doing an isometric motor task and vocalizing. Readers should be made a bit more aware that these tasks are different, vocal fold vibrations are the product of a complex interplay between a transglottal pressure difference, vocal fold stiffness, and Bernoulli forces. Individual differences may also play a role (how large are vocal folds, are false vocal folds coupled to the vocal folds in these tasks etc.).

Methodology needs additional information for clarity
- I think using voice quality measures over an entire sentence deserves some explanation. Voice quality is highly dependent on the phoneme/segment produced – it can only be measured when f0 is present, but not in voiceless sounds where the glottis is either open or closed, but vocal folds are stiff. I see that participants always realized the same sentence, in this sense the data always include the same segments (I hope there is no reduction), but I feel some explanations for the current approach are needed.
- You may also provide a sentence motivating why you use this large number of measures. I am aware that there are many. On the other hand, some may be correlated or redundant. Is this a rather exploratory analysis or are the selected measures motivated? If so, how?
- Vocal tremor has been introduced in line with vocal effort. However, I went back and forth in the text to see based on which measure the tremor was measured. Including the term tremor in the part on frequency modulation might be advisable, because then this is clear (right now two terms are used). Since it occurs in certain conditions, why is it then added to the stimuli for the perception experiment? And why is it added even to the control condition (to a minor extent)? If it does not occur in some conditions, why should an additional tremor be included in the perception experiment?
- The acoustic data gave me the impression that some of the files are recorded with the close and some with the more distant microphone (tl and cl in the name of the file) – the 2 mics are also mentioned in the methods section. How did you deal with this in the analyses? It may not affect f0 measures, but the intensity and HNR measures. Can you please clarify?
- Would be nice to explain random forests in a few sentences.

Check definition
The definition of linguistic prosody (Line 125) should receive a reference or be rephrased because it differs from what is commonly known as prosody (at least to me). I have never heard that prosody is defined with respect to consciousness (because consciousness is hard to define) and relates only to f0. Some prosodists in North America call f0 modulations intonation, but not prosody.
Here might be a common definition which at least several colleagues and I agree with: “The linguistic term prosody refers to the suprasegmental structure of the utterance, encoding prominence and phrasal organization [4 –7].” (page 1) in Krivokapić, J. (2014). Gestural coordination at prosodic boundaries and its role for prosodic structure and speech planning processes. Philosophical Transactions of the Royal Society B: Biological Sciences, 369(1658), 20130397.
See also other references: https://www.tandfonline.com/doi/abs/10.1080/01690961003589492?journalCode=plcp20

Experimental design

Research on effortful motions and their impact on speech parameters is limited. The author's work is original, since it focuses on an isometric task, while most of the previous work is on endurance tasks. The first experiment is well-designed with 4 different tasks. The results are limited to the speech material of one sentence, so they deserve further testing before they can be generalized.
The perception experiment is also well designed and clear.
I had some remarks on adding tremor - which I included in the section on basic reporting.

Validity of the findings

The author provided the acoustic data and scripts, i.e. the data can be replicated.
It would be good, if some reference could additionally be included showing a link between physical fitness and isometric motor tasks. Alternatively, the author may state that this link is an assumption.
The results are discussed, but I am afraid, I miss a clear stated conclusion.

Additional comments

Remark:
I don’t have the feeling that the consent form for the participants is typical. In the upper part, several pieces of information are missing concerning personal data protection, data storage, etc. One gives consent about data storage without having information about this earlier, and the ethics was approved in France while the experiment is in Sweden. Since the regulations about ethics can be different in each country, I won’t judge this.
Personally, I found: https://www.youtube.com/watch?v=nmJKY59NX8o) pretty violent and was not prepared to watch this when clicking on the link. Just wanted to let you know.
I am also not sure whether vocalization during copulation is something to introduce in this context (lines 421-425). It has nothing to do with isometric muscle activation as far as I can tell. Moreover, my experience may be limited, but I think we should keep the potential “WEIRD” people (Henrich et al.'s famous paper) bias in mind. I was very often surprised by how loud females are in several hotels in France. I did some fieldwork on an island in the Pacific and lived in villages with houses that have walls like paper where I heard zero. I would assume a cultural bias here, which may also depend on the function of marriage or being a couple.

Some further suggestions:
Line 30: I think the following paper might make a lot of sense to be included at this point – it even has a related title:
Wright, E., Grawunder, S., Ndayishimiye, E., Galbany, J., McFarlin, S. C., Stoinski, T. S., & Robbins, M. M. (2021). Chest beats as an honest signal of body size in male mountain gorillas (Gorilla beringei beringei). Scientific reports, 11(1), 1-8.

·

Basic reporting

In this work, Andrey carried out 3 studies to examine the vocal characteristics of genuine physical exertion. First participants were asked to say the same sentence while performing an L-sit, first without any voice-related instructions and then asked to either conceal or imitate their effort without actually performing the exercise. Along with an independent analysis of the results in order to characterize the differences between genuine and non-genuine sound of effort exertion, these recordings were then sent to other sets of participants in two follow-up studies to rate the amount of effort they think the original participants experienced. These recordings were added a tremor at a certain time to test its unique contribution to effort communication. The results indicated that the conditions can be differentiated by their vocal characteristics in the first study and that participants' ratings were affected by the effort manipulation.

Experimental design

This work answers all criteria of experimental design.
Abstract:
Line 20: Please elaborate on what "vocal effort" is or how it is different from other non-effort vocal properties. In addition, after reading the rest of the manuscript I understand there were actually 3 separate studies conducted. I think this can be clearly stated in the abstract along with the sample size of each so readers can easily evaluate the scale of this work.
Intro:
Very well written and interesting
Methods:
Were participants paid?
For how long was the L-sit exercise? If possible, it might be clearer if the author can provide an actual picture of a model performing the exercise along with the drawings so readers from sports science fields can better assess the settings when trying to build upon this work for future experiments.
Please elaborate on why these variables were chosen for analysis. If the reason is for exploratory purposes for some or all it's also fine.
There are two additional studies described under "Perceptual tests". I think this work is also relevant to the topic and very interesting subjectively – I suggest it will be better if these will be mentioned in the abstract and introduction as well.
Although absolutely not mandatory, I think it might be clearer if the manuscript would have taken the form of "Study 1", "Study 2" etc. to clarify the specific independent contribution of each study.

Validity of the findings

This work answers all criteria of validity.

Results
Line 247-253: please add exact statistics to clarify of the results
Figure 2: please indicate specifically in the captions what the x- and y-axes mean
Line 262: Please delete the "and"
Please make the OSF link to the data available in the results section
Discussion
I think it is very well written and insightful, I also enjoyed the reference in line 348.

Aviv Emanuel

Additional comments

I thoroughly enjoyed reading this manuscript. It was well written and easy to follow, making it a pleasure to review. The studies presents important insights into the topic at hand. Overall, this manuscript makes a valuable contribution to the field and I believe it will be of great interest to others working in this area. I have some minor-medium suggestions that I think can aid the clarity of this manuscript and hence increase its spread and citation count. From my perspective, the author is free to choose whether to follow these suggestions or not without any consequences for the publication of this manuscript.
My suggestions are written by section along with the requirements of PeerJ,

·

Basic reporting

I think the reporting of the relevant literature is formidable. There is wide breath of relevant studies cited.
Maybe I want to see a little more reflection on the potential significance of concealment skills with respect to some other related research. For example, there is some research suggesting that allometry breaking is indicative of communication complexity. E.g., birds generally scale the duration of their vocal units with their wingbeats, likely due to biomechanical constraints. Coincedentally, the subset of birds that are vocal learners, have a less strong coupling with wingbeats, suggesting that they can somehow escape allometric scaling relations:
Berg, K. S., Delgado, S., & Mata-Betancourt, A. (2019). Phylogenetic and kinematic constraints on avian flight signals. Proceedings of the Royal Society B, 286(1911), 20191083.
A similar point has been made on the relation of vocal tract properties and acoustics:
Garcia, M., & Ravignani, A. (2020). Acoustic allometry and vocal learning in mammals. Biology Letters, 16: 20200081. doi:10.1098/rsbl.2020.0081.
In line of these studies, I wonder whether the authors are interested to delve a little deeper in the significance of indexical cues and/or animals faking them. We should always wonder, if something can be faked, it must lose its indexical value. I think a well-known book that the authors might want skim is:
A. Zahavi, A. Zahavi
The Handicap Principle: A Missing Piece of Darwin's Puzzle
Oxford University Press (1999)

Experimental design

I think the analyses procedure, experimental design, statistical analyses, and motivations are really well done. I dont see any issues that should preclude publication.

Validity of the findings

A general reflection I missed is on the concealment and pretending mechanism. How do we know that some of the physical effects of the L-sit are either invoked (as in ‘pretending’) or counteracted with antagonistic muscular adjustments (as in ‘concealment’). In this sense the “pretending” of physical effort, is still very much the use of physical effort by for example tensing ones rectus abdominus all the same, rather than some kind of top-down cognitive ‘pretending’ using some more subtle control mechanisms of the vocal system. Perhaps, it is a mixture of course, where some kind of acoustic cues can be induced by invoking some self-generated muscle activations that still is a form of physical effort, whereas others are induced by less physically effortful modulations (e.g., vibrato).

---

## Round 0.2 · accepted · Accept

Thank you for the detailed response letter and revised submission. We are delighted to accept your manuscript for publication.

·

Basic reporting

The basic reporting of the paper is clear. It covers an interdisciplinary area and deals with vocal effort produced in an isometric hold and its perception in two different studies. The studies are original and contribute to ongoing investigations on links between physical effort and the voice.
The article is professionally written, data and scripts are shared and the hypotheses are clearly defined.

Experimental design

The experimental design, methodological procedures, and processing steps are clear and the author did an effort to clarify all outstanding questions in the revisions.

Validity of the findings

'no comment'

Additional comments

Thank you for answering my concerns in detail. Looking forward to seeing this work published.
Line 201: prosodic measures - > acoustic measures? (you mentioned that you removed the word prosody)